# Combined Effects of Myeloid Cells in the Neuroblastoma Tumor Microenvironment

**DOI:** 10.3390/cancers13071743

**Published:** 2021-04-06

**Authors:** Jennifer Frosch, Ilia Leontari, John Anderson

**Affiliations:** UCL Institute of Child Health, Developmental Biology and Cancer Section, University College London, London WC1N 1EH, UK; j.frosch@ucl.ac.uk (J.F.); ilia.leontari.20@ucl.ac.uk (I.L.)

**Keywords:** neuroblastoma, tumor microenvironment, myeloid-derived suppressor cells, tumor associated macrophages, myeloid cells, immunotherapy

## Abstract

**Simple Summary:**

Neuroblastoma is a type of childhood solid cancer often characterized by resistance to treatment and with poor prognosis. The tumors contain a mixture of cell types as well as cancer cells, and one of the most common is a type of blood cells called myeloid cells. In this review article we present a summary of what is known about the function of myeloid cells within neuroblastoma tumors.

**Abstract:**

Despite multimodal treatment, survival chances for high-risk neuroblastoma patients remain poor. Immunotherapeutic approaches focusing on the activation and/or modification of host immunity for eliminating tumor cells, such as chimeric antigen receptor (CAR) T cells, are currently in development, however clinical trials have failed to reproduce the preclinical results. The tumor microenvironment is emerging as a major contributor to immune suppression and tumor evasion in solid cancers and thus has to be overcome for therapies relying on a functional immune response. Among the cellular components of the neuroblastoma tumor microenvironment, suppressive myeloid cells have been described as key players in inhibition of antitumor immune responses and have been shown to positively correlate with more aggressive disease, resistance to treatments, and overall poor prognosis. This review article summarizes how neuroblastoma-driven inflammation induces suppressive myeloid cells in the tumor microenvironment and how they in turn sustain the tumor niche through suppressor functions, such as nutrient depletion and generation of oxidative stress. Numerous preclinical studies have suggested a range of drug and cellular therapy approaches to overcome myeloid-derived suppression in neuroblastoma that warrant evaluation in future clinical studies.

## 1. Introduction

As one of the more recently introduced treatments in neuroblastoma, immunotherapy targeting tumors via the abundantly expressed tumor-associated antigen disialoganglioside GD2 has shown promising results [1]. While consolidative therapy with anti-GD2 monoclonal antibodies (mAb), such as dinutuximab, has already been integrated into standard protocols for high-risk patients, ongoing clinical studies investigating chimeric antigen receptor (CAR) T cells in neuroblastoma have revealed limited responses despite preclinical reports of efficacy [2]. Demonstrated issues included minimal CAR T expansion as well as reduced immune activation when targeting GD2 with CAR T cells [3,4,5,6], suggesting the influence of an immunosuppressive mechanism at play.

One of the reasons neuroblastoma is hard to treat with immunity-based approaches is its inflammatory character, which leads to the formation of a highly immunosuppressive microenvironment consisting of soluble factors, stromal cells, and alternatively activated immune effectors. Studies investigating neuroblastoma in transgenic TH-MYCN mouse models have shown that tumors undergo a transition from early lesions highly infiltrated with lymphocytes to progressively increasing numbers of myeloid populations in advanced tumors [7,8]. This was further confirmed in immunohistochemical analyses of human tumor samples, where neuroblastoma and other pediatric solid tumors were heavily infiltrated with Cluster of differentiation (CD) 68+ myeloid cells [9]. We now know that these myeloid populations predominantly consist of type 2 tumor-associated macrophages (TAM) and myeloid-derived suppressor cells (MDSC) [10,11,12]. This is reflected in a 14-gene signature associated with poor prognosis in neuroblastoma, which contains five genes related to myeloid populations and their suppressive mechanisms (CD14, CD33, IL-10, CD16, IL-6R) [10].

Therefore, detailed understanding of the induction and suppressive mechanisms of these tumor microenvironment (TME)-associated myeloid populations in solid malignancies, including neuroblastoma, is crucial for the design of novel immunotherapeutic treatments and combinatorial approaches. This review will discuss cellular components of the immunosuppressive microenvironment of neuroblastoma, with a focus on TAM- and MDSC-mediated suppression, and how these myeloid populations can be targeted in the immunotherapy of neuroblastoma.

## 2. The Suppressive Tumor Microenvironment of Neuroblastoma

Neuroblastoma is characterized by its complex immunosuppressive microenvironment. The combined effects of tumor cells, stromal components, and infiltrating immune cells leads to an inflammatory, hypoxic, vascular, and angiogenic niche that supports tumorigenesis and immune evasion [13,14,15]. Inhibitory immune effectors like TAMs, MDSCs, and Treg cells infiltrate tumors as the lesion progresses, contributing to increasingly immunosuppressive sites [8]. Indeed, Asgharzadeh et al. have identified 14 genes associated with poor outcome in metastatic neuroblastoma, of which 5 were inflammation-related genes and 9 were tumor-related genes [10].

Defects in the antigen presenting machinery (APM) and low levels of major histocompatibility complex (MHC) I molecules in neuroblastoma hinder engagement with cytotoxic T cells and their subsequent activation [16,17,18]. T cell function is further inhibited through secretion of soluble factors like transforming growth factor-β (TGF-β), interleukin (IL)-10, and galectin-1 [15,19,20,21]. Moreover, MYCN-dependent downregulation of NKG2D ligands, such as MIC-A, MIC-B, ULBP-1, ULBP-2, and ULBP-3, as well as release of their soluble forms support evasion of innate immune responses, including NK cell activation [22,23,24]. Studies in both murine and human neuroblastoma have demonstrated that the tumor cells can also suppress immunity through arginase (Arg)-2 activity [25]. The resulting depletion of L-arginine leads to downregulation of the CD3z chain on T cells, which induces proliferative arrest of the cells, as well as further contributes to the induction of immunosuppressive phenotypes in the tumor niche [25,26].

Abundantly found in the tumor stroma, mesenchymal stem/stromal cells (MSC) and cancer-associated fibroblasts (CAF) strongly support tumor survival through their angiogenic and immunosuppressive effects. It has been reported in many tumors that MSCs can promote cell invasiveness, migration, and endothelial-to-mesenchymal transition (EMT) of cancer cells. Indeed, studies have found that MSCs isolated from neuroblastoma tumors carried transcriptomic profiles associated with EMT, which in turn were associated with poor prognosis, and expressed high levels of C-X-C Motif Chemokine Ligand 12 (CXCL12), promoting neuroblastoma metastasis and invasiveness [27,28,29]. MSCs also protect the tumor from oxidative stress through the production and activation of antioxidant enzymes against hydrogen peroxide in the TME [30]. In neuroblastoma, CAFs were first recognized as α-smooth muscle actin (α-SMA)-positive cells, which were associated with an increase in microvascular proliferation and poor clinical outcome [31,32]. CAFs are activated by soluble factors in the tumor proximity, such as TGF-β, and in turn release abundant tumor-promoting and angiogenic factors, thus contributing to the establishment of a refractory TME, tumor growth, and disease progression [33]. CAFs promote vascular proliferation through the angiogenic factors, vascular endothelial growth factor A (VEGF-A) and IL-8, and suppress NK and T cells through secretion of anti-inflammatory cytokines like TGF-β and IL-6 [34,35]. Furthermore, the produced factors activate signal transducer and activator of transcription (STAT)3 and extracellular regulated kinase (ERK 1/2)-mediated signaling in neuroblastoma cells, supporting proliferation, survival, and chemokine resistance of the tumor [36]. Like MSCs, CAFs also release chemokines, such as C-C Motif Chemokine Ligand 2 (CCL2), to recruit further infiltrating immune cells to the TME. Indeed, it has been reported that in neuroblastoma, CAFs and TAMs are often in close proximity with reciprocal functionality [32,37]. While CAFs recruit TAMs via CCL2 and shift differentiation towards the M2 phenotype via production of prostaglandin E 2 (PGE2) [38,39], TAMs in turn enhance CAF-mediated STAT3 activation through release of the soluble agonistic IL-6 receptor (IL-6R), and promote CAF invasiveness and proliferation [30,36]. Moreover, Hashimoto et al. have reported that the number of TAMs and the area of CAFs in neuroblastoma tumors were positively correlated with clinical stage and MYCN amplification [37].

In the solid tumor TME, infiltrating T cells can acquire an immunoregulatory phenotype. Regulatory T cells (Tregs) suppress antitumor immunity via pleiotropic inhibitory mechanisms and thus support immune evasion. Indeed, in neuroblastoma mouse models, depletion of Tregs increased the efficacy of T cell-mediated immunotherapy [40,41]. Tregs contribute to the immunosuppressive milieu by secreting inhibitory cytokines, such as IL-10, TGF-β, and IL-35, and disrupt maturation of antigen-presenting cells (APC) through CTLA-4 engagement. Furthermore, Tregs express granzyme and perforin, which mediate cytotoxic effects on effector T cells [33,42,43]. Through their expression of high-affinity receptors for important inflammatory cytokines, such as IL-2, IL-7, IL-12, and IL-15, Tregs act as competitive sinks and further suppress function and activation of cytotoxic T cells [44,45,46].

The local accumulation of C-C motif chemokines, such as CCL2, CCL7, and CXCL1, as well as the hypoxic milieu of neuroblastoma, results in the recruitment of high numbers of myeloid cells [37,47]. As a result of the local availability of cytokines and growth factors, myeloid differentiation is often disrupted in solid tumors, which leads to the generation of heterogeneous immature populations with suppressive functions, generally described as MDSC [48,49]. Moreover, the populations that do undergo differentiation in the solid TME acquire suppressive phenotypes and disrupted functions. Dendritic cells in the TME demonstrate defective antigen presentation function caused by the local effects of cytokines and growth factors, such as IL-6, IL-10, and VEGF [44], and downregulate both MHC and costimulatory receptor expression on their surfaces, resulting in less T cell priming and activation [50]. Studies in solid tumors show that hypoxic factors in the TME, such as hypoxia inducible factor 1α (HIF1α), promote differentiation of MDSCs into TAMs, further contributing to the range of immunosuppressive cell populations [47,51]. Considering their wide array of suppressive mechanisms, their prognostic potential, and their abundance in the neuroblastoma environment, suppressive myeloid populations, including MDSCs and TAMs, present an interesting target for improving immunotherapy of solid tumors. The next sections therefore will focus on their role in cancer progression and immune suppression, and how preclinical studies are attempting to overcome myeloid-mediated suppression in neuroblastoma.

## 3. Myeloid Cell Populations in Neuroblastoma

### 3.1. Myeloid Heterogeneity

In the TME, cells of myeloid origin are of heterogeneous character and are generally of an immature and immunosuppressive phenotype. Due to the transitional expression of surface markers, phenotypes of the different subsets are difficult to dissect and often overlap (Figure 1). In mice, MDSC can be defined by the markers of Gr-1, Ly6C, and Ly6G, and are discriminated from TAM via the marker F4/80 [52]. In humans, the distinction between the two myeloid subsets is less clear. While MDSC are generally considered immature myeloid cells with low expression of class II MHC, and which do not express the macrophage marker CD68 [53], the marker is not as heterogeneously expressed among macrophages as once thought and may thus lead to false categorization as MDSC. Another marker classically used for the characterization of suppressive macrophages is CD163, but recent studies have found that CD163 can be detected on a variety of human anti-inflammatory myeloid cells, as it is upregulated by IL-10, glucocorticoids, and M-CSF, and has been found on tumor-infiltrating cells with negligible expression of CD68 [54,55,56,57]. Furthermore, in acute liver failure syndromes, a subset of circulating MDSC with anti-inflammatory functions has also been found to express CD163 [58]. The expression of markers considered specific for MDSC and TAM seems to vary [59] and underlines that further efforts have to be made into characterizing subsets correctly, such as recently proposed immunofluorescent multiparameter assays [56].

#### 3.1.1. Tumor-Associated Macrophages

Macrophages are a class of heterogeneous immune cells of myeloid origin which primarily exert phagocytic functions in immune reactions and tissue remodeling [60]. Depending on environmental stimuli, macrophage polarization can result in two main phenotypes: M1 and M2. While M1 macrophages are characterized by their immunogenic functions and are typically activated by immunostimulatory signals lipopolysaccharide (LPS) or interferon gamma (IFN-γ), the M2 phenotype is stimulated by IL-4 and IL-13 and plays an immunoregulatory role in wound healing and tissue remodeling, exerted through the secretion of VEGF, matrix metalloproteinase 9 (MMP9), IL-10, and TGF-β [61,62]. M1 and M2 macrophages do not only differ in their functions but can also be distinguished by their genetic profile [63,64], metabolism [65,66], and epigenetic modifications [67]. Macrophages with M2 polarization can be further characterized into subtypes with differing profiles (Figure 1). M2a macrophages are mainly involved in wound healing through secretion of TGF-β and other factors. Stimulated by toll-like receptor (TLR) or interleukin 1 receptor (IL-1R) agonists, M2b macrophages exert immunoregulatory functions via release of cytokines, such as IL-10, in cancer and other diseases. Lastly, M2c macrophages are involved in tissue remodeling and dampen the immune response through the release of TGF-β and IL-10 [62,68,69]. Importantly, macrophages are a highly heterogeneous and fluid population, which can easily transition between different phenotypes depending on environmental stimuli [70]. The binary M1 and M2 classification is an oversimplification, however due to the current limitations in clearly differentiating subtypes and activation states, it will be used in the subsequent sections of this review.

Macrophages in the TME are collectively described as TAMs and typically mirror the M2 phenotype. Interestingly, their presence has been found to correlate with worse prognosis in neuroblastoma and other solid tumors [10,71]. Furthermore, studies suggest that TAMs can promote tumor progression by upregulating oncogenic MYC expression through activation of the STAT3 pathway in non-MYCN amplified neuroblastoma cells [72].

Expressing M2 macrophage markers like CD163 and CD206 (Table 1), TAMs suppress the immune response through the secretion of regulatory cytokines, such as IL-10 and TGF-β, and promote tumorigenesis via factors like VEGF and MMPs [73]. Within tumors, distinct subpopulations of TAMs can be found, depending on their location, with varying degrees of M2-like character and functions. For example, perivascular TAMs contribute to metastasis, while TAMs at the tumor–stroma interface promote angiogenesis [74,75]. Recent studies in lung carcinoma and other solid tumors have found that hypoxia within the TME promotes differentiation of MDSC to TAMs and shapes their phenotype towards lowered hypoxia-sensitive gene expression and angiogenic activity but does not change the M2-like character [51,76]. This heterogeneity among TAMs within tumors is often overlooked in preclinical studies, which tend to oversimplify TAMs as a single homogeneous population, limiting the informative value of their results [62].

#### 3.1.2. Myeloid-Derived Suppressor Cells

MDSC, a heterogeneous population of immune cells with myeloid lineage, have been known to exist in cancer for over 30 years, but only recently have they caught more attention towards their critical function in shaping the TME [77,78]. Santilli et al. have demonstrated that co-injecting immunocompetent mice with neuroblastoma cells and MDSC greatly increases tumor growth [11]. Furthermore, studies on rhabdomyosarcoma in mice have shown that disruption of CXCR2-dependent MDSC trafficking to the tumor resulted in significantly increased antitumor effects [79]. MDSC can also be found under normal physiological conditions, where they regulate immune tolerance. For example, MDSC mediate maternal–fetal tolerance in pregnancy and have also been found to accumulate with age [80,81,82]. In cancer, the major factor driving MDSC accumulation is the chronic, low-grade inflammation of the TME, where the alternatively activated myeloid cells act through a wide range of mechanisms mediating tumor progression and immune inhibition [83].

Generally, MDSC in cancer describes immature cells of myeloid origin with immunosuppressive character [77]. They have been observed both in mice and in human disease and can be grouped into at least two subpopulations: mononuclear MDSC (M-MDSC) and granulocytic or polymorphonuclear MDSC (PMN-MDSC). Correspondingly, M-MDSC are morphologically similar to mononuclear blood monocytes, while PMN-MDSC resemble polymorphonuclear neutrophils [53]. All MDSC in mouse and human are negative for T cell, B cell, and NK cell lineage markers [78] (Table 1). In mice, MDSC can be characterized by their surface expression of CD11b combined with Ly6C or Ly6G, which are known collectively as Gr-1. Mouse M-MDSC are CD11b+ Ly6C+ Ly6G−/low, whereas PMN-MDSC are CD11b+ Ly6G+ [84,85]. In human, there are no MDSC-specific markers, so that they can only be distinguished using a combination of common immune markers. Human M-MDSC are CD33+ CD11b+ CD14+ CD15− HLA-DR−/low and PMN-MDSC are CD33dim CD11b+ CD15+ CD14− HLA-DR−/low. Furthermore, there is a third population described in human, early-stage MDSC (eMDSC), which are CD33+ CD11b+ HLA-DR−/low but lack myeloid lineage markers (CD14, CD15, CD66b) [53,86,87]. While M-MDSC can be phenotypically distinguished from healthy monocytes by absence or low expression of HLA-DR, there has been an ongoing uncertainty about the differences between PMN-MDSC and neutrophils [88,89]. One way to separate the two populations is by standard Ficoll density gradients (1.077 g/L), where neutrophils band at a higher density than PMN-MDSC, albeit some activated neutrophils may be included in the low-density fraction [90]. More recent advances in whole-genome analysis and flow cytometry of samples from head and neck cancer patients have revealed the lectin-type oxidized low-density lipoprotein receptor 1 (LOX1) as a marker found in human PMN-MDSC, but not in neutrophils [91]. In mice, Cd84 and junctional adhesion molecule have been discussed as markers distinguishing PMN-MDSC from neutrophils [92].

### 3.2. Contribution of Myeloid Cells to the Tumor Microenvironment

#### 3.2.1. The Role of TAMs

Recruited by chemoattractants including CCL2, CCL20, CXCL12, and CSF-1 [37,93,94,95], macrophages in the proximity of solid tumors are quickly reprogrammed towards an M2-like phenotype in a process driven by the local hypoxia [96]. The polarized TAMs then contribute to tumor progression, angiogenesis, and immunosuppression by expressing factors such as IL-6, VEGF, inducible nitric oxide synthase (iNOS), and arginase [62,96]. TAMs have been characterized in a range of solid tumors, which can provide indications for their contributions in neuroblastoma tumorigenesis.

##### Contribution to Tumor Progression

Progression of neuroblastoma and other solid tumors is promoted by TAMs through the secretion of factors like VEGF, Platelet-derived growth factor (PDGF), TGF-β, MMP2, and MMP9, which lead to extracellular matrix remodeling and increased neoangiogenesis [97,98] (Figure 2). It has been demonstrated that the pro-angiogenic niche in tumors result from the combined effects of TAMs and local hypoxia, and that in neuroblastoma, HIF2a and TAMs are markers of the perivascular niche [99,100]. The upregulated levels of HIF1α/2α in TAMs further promote angiogenesis through inducing expression of pro-angiogenic factors, including VEGF, TNFα, and MMP [93,96]. These factors released by TAMs also contribute to tumor metastasis and facilitate both the tumor’s release from the primary site and the establishment at secondary sites by increasing vessel permeability [73]. Furthermore, Hashimoto et al. have shown that in neuroblastoma, TAMs enhance tumor invasion through increased expression of CXCL2 and interactions with its receptor CXCR2 on neuroblastoma cells [37].

##### Contribution to the Immunosuppressive Microenvironment

TAMs contribute to the immune suppression in the TME both through stimulation of other suppressive populations, such as Tregs and MDSC, and through direct inhibition of effectors including NK cells and cytotoxic T cells [62] (Figure 2).

Studies in neuroblastoma have demonstrated that tumor cells induce suppressive macrophage populations, which inhibit effector NK cells via secretion of IL-6 and TGF-β and thus dampen the efficiency of anti-GD2-targeted immunotherapy [19,101]. TAMs also exert a suppressive effect on infiltrating cytotoxic T cells, which is mediated both by their release of inhibitory cytokines as well as through direct interactions with T cell surface receptors. In human lung squamous-cell carcinomas, melanoma, and other solid tumors, TAMs simulate APM interactions and thus prevent the T cell effectors from binding to functional APCs, while activating checkpoint blockade through engagement with inhibitory co-receptors PD-1 and CTLA-4 [102,103,104]. Moreover, TAMs express the enzymes Arg-1 and indoleamine 2,3-dioxygenase (IDO)-1/2, leading to a local metabolism of L-arginine and L-tryptophan respectively, which are essential for T cell activation [105]. Through their release of soluble factors, TAMs actively shape the immunosuppressive character of the TME by affecting other suppressive populations. Indeed, TAM-secreted CCL2 recruits Tregs, which in turn promote differentiation of M2 macrophages and exert their immunosuppressive functions via IL-10 and TGF-β release [105,106]. In prostate carcinoma, the secreted TGF-β also mediates recruitment and enhancement of CAFs in the TME by facilitating mesenchymal-to-epithelial transition (MET) [107]. This increase in local CAFs leads to higher levels of stromal cell-derived factor 1 (SDF1/CXCL12), thus further enforcing infiltration of TAMs and endothelial progenitor cells to promote angiogenesis [37,95,108]. Via the release of CCL2, TAMs recruit further myeloid populations to the tumor, and through the released factors IL-10 and TGF-β, mitigate induction of immunosuppressive MDSC and suppression of APC recruitment and function [109]. In turn, the MDSC produce more IL-10 and can quickly differentiate into M2-like macrophages, providing a positive feedback towards more immunosuppression, as shown in several solid tumor models [51] (Figure 2).

#### 3.2.2. The Role of MDSC

There is only limited data available on the clinical significance of MDSC in neuroblastoma. Studies in neuroblastoma mouse models have reported the accumulation of MDSC during tumor progression and that MDSC-released factors including Reactive oxygen species (ROS)-, Arg-1-, and TGF-β-mediated tumor growth [7,110]. Indeed, Santilli et al. observed a drastic increase of immunosuppressive MDSC-like cells in the blood and tumors of neuroblastoma patients [11].

The predominant population in tumor-bearing mice and cancer patients are often PMN-MDSC, however there are differences between cancer types and inconclusive reports about which population is the most suppressive [49,78]. It has been shown in mice that M-MDSC can be considered more immune suppressive than PMN-MDSC on a per-cell basis. Studies in patients with head and neck and urothelial cancer, however, have demonstrated a more suppressive per-cell activity of the PMN-MDSC populations [86]. To our knowledge, there are no studies comparing the suppressive potential of MDSC subsets in neuroblastoma, so the predominant suppressive subset is not known.

##### Recruitment and Activation of MDSC

During hematopoiesis, hematopoietic stem cells (HSC) in the bone marrow differentiate into common myeloid progenitor cells (CMPs), which further give rise to granulocytes, DC, and macrophages. When exposed to stress conditions, such as cancer-associated inflammation, this healthy differentiation process is perturbed, leading to the accumulation of immature myeloid cells during so-called emergency myelopoiesis. Excessive levels of tumor-produced granulocyte colony-stimulating factor (G-CSF) and granulocyte-macrophage colony-stimulating factor (GM-CSF) cause the faulty myelopoiesis through downregulation of the transcription factor interferon regulatory factor 8 (IRF8) and alternatively activate these MDSC, equipping them with potent immunosuppressive activity [111,112,113].

The accumulation of MDSC in tumor-bearing individuals has been shown to depend on the inflammatory mediators IL-1β, IL-6, and Prostaglandin-E2 (PGE2) [114,115,116,117,118]. A crucial pathway for this is via the key MDSC activator STAT3. Activation of STAT3 induces upregulation of p47phox and gp91, which in turn increases the production of reactive oxygen species by MDSC, as shown in subcutaneous tumor models in mice [119,120]. STAT3 activation in MDSC is facilitated by G-CSF and GM-CSF [121,122,123,124], hypoxia in the TME [51], and IL-1β and IL-6 [114,115,125,126].

Another activator of MDSC is the highly proinflammatory alarmin S100A8/A9 complex, which has been demonstrated to activate Nuclear Factor Kappa B (NF-κB) and STAT3 pathways by binding TLR4 and carboxylated N-glycans on the receptor for advanced glycation end products (RAGE) on MDSC in colon carcinoma-bearing mice [78,127]. Sinha et al. have also demonstrated that S100A8/A9 acts as a chemoattractant for MDSC in solid tumors and that MDSC secrete S100A8 and S100A9 in the serum of tumor-bearing mice, thus creating an autocrine feedback loop [128]. Given its impact on MDSC accumulation and activation, S100A8/A9 has been suggested as a novel biomarker for human MDSC [129], which is yet to be elucidated in neuroblastoma.

##### Promotion of Tumor Progression

MDSC not only exert significant immunosuppressive effects but can also promote tumor progression through non-immunogenic mechanisms (Figure 2). Studies in tumor-bearing mice and patients with triple-negative breast cancer demonstrated MDSC-mediated induction of angiogenesis and metastasis through the production of Mmp9 [130,131]. The protein promotes neo-angiogenesis through increasing VEGF release and facilitates invasion of tumor cells through the degradation of the extracellular matrix of healthy tissue. Metastasis is further supported through increased epithelial–mesenchymal transition induced by MDSC, which has been reported to be dependent on TGF-β, Epidermal growh factor and hepatocyte growth factor (EGF, and HGF) signaling pathways in a melanoma mouse model [132]. In mice with lung adenomas and adenocarcinomas, MDSC protect tumor cells from senescence by secreting IL-1 receptor agonist, thus blocking the production of pro-senescence factors and promoting tumor growth [133].

##### Contribution to the Immunosuppressive Microenvironment

Early studies in breast cancer models revealed that MDSC exert their immunological functions through the release of soluble mediators, which rely on cell contact because of their short half-life and concentration [134]. More recent investigations have additionally demonstrated the existence of MDSC-derived exosomes with suppressive activity and the ability to shape other suppressive populations in the TME [135,136,137]. MDSC influence a wide range of immune cells within the TME through their heterogeneous mechanisms. While T cells are their key target, with suppression of T cell proliferation or cytokine release considered one of the hallmarks of defining MDSC [53], other effectors of immunity, including NK cells, APCs, Tregs, and B cells, have also been described to be targeted by MDSC-mediated measures (Figure 2). Importantly, although MDSC possess a multitude of mechanisms to interfere with pathways of the immune system, myeloid populations in the TME are heterogeneous and the predominant mechanisms may change depending on differentiation stage, extracellular milieu, and cellular composition of tumors [78].

MDSC suppress T cells through (1) limiting amino acid availability, (2) disruption of T cell functions through production of reactive species, and (3) prevention of trafficking. A key MDSC marker, Arg-1, attenuates T cell activation through depletion of L-arginine, which is essential for manufacture of the T cell receptor (TCR) ζ chain [138,139,140]. Furthermore, MDSC reduce the availability of L-tryptophan by expressing IDO [141] and decrease extracellular levels of the essential amino acid cysteine by hoarding the crucial building block cystine and suppressing cysteine release in APCs [142]. Alongside arg-1, reactive oxygen/nitrogen species (ROS/RNS) production is found in both human and mouse MDSC and therefore serves as a MDSC biomarker [53]. MDSC subsets can generate both ROS and RNS, of which ROS are predominantly generated by PMN-MDSC [49,143]. ROS production involves phosphorylation of STAT3, which in turn upregulates expression of p47phox and gp91, two subunits of the ROS-generating NADPH oxidase. This facilitates superoxide to react with nitric oxide (NO), generated by iNOS-mediated breakdown of L-arginine, to produce peroxynitrite (PNT) [53,120,144]. PNT inhibits T cell activation through nitration of the TCR as well as MHC nitration on APCs, preventing antigen recognition [145,146]. Moreover, PNT nitrates chemokines important for T cell recruitment and infiltration [147]. Studies in breast cancer have shown that MDSC also prevent T cell activation in the lymph node through expression of the enzyme A disintegrin and metalloprotease 17 (ADAM-17), which cleaves L-selectin, a crucial surface molecule for T cell extravasation from the blood and lymphatics into the lymph nodes [148,149]. Furthermore, in hepatocellular carcinoma, MDSC disrupt the T cell stimulatory activity of DCs through secretion of IL-10, which inhibits TLR-ligand-induced IL-12 production of DCs [150]. Studies in mice models of metastatic colon cancer have demonstrated that MDSC also mediate the development of immune-suppressive Tregs through CD40–CD40L interaction in an IFN-γ- and IL-10-dependent process [151,152]. In mouse and human carcinomas, MDSC-mediated suppression of NK cell functions is cell contact-dependent and mainly exerted via engagement with NKp30 on NK cells [153,154]. In mice, a novel Ly6C-negative MDSC subset has been reported to target NK cells after activation by tumor-released IL-1β [154,155], whereas in melanoma patients, M-MDSC are activated by PGE2 and impair NK cell-mediated cytotoxicity through the release of TGF-β [117].

Lastly, the predominant myeloid populations within the TME work together in sustaining the suppressive niche through a feedback-loop dependent on IL-1β and MDSC-expressed TLR4. The crosstalk between MDSC and macrophages enhances TAM protumor activity: MDSC inhibit antitumor M1-like macrophages by secretion of IL-10 and induce downregulation of MHC class II on macrophages, thus attenuating M2-like phenotypes. In turn, macrophages promote MDSC production of IL-10, creating a positive feedback loop towards the creation of TAM [156,157,158]. Furthermore, when entering hypoxic regions of solid tumors, M-MDSC rapidly differentiate to TAM [51].

## 4. Therapeutic Strategies to Target Suppressive Myeloid Cells

The potent immunoregulatory microenvironment of solid cancers, like neuroblastoma, poses an important hurdle to overcome in immunotherapy. With their heterogeneous range of suppressive mechanisms, myeloid cells have become a major target to consider for combinatorial approaches with CAR T cells, checkpoint inhibitors, and NK cell-based therapies. Indeed, it has been demonstrated that not only do myeloid phenotypes positively correlate with CAR T therapy failure [159], but also that ablation of these populations can improve antitumor effects of various immunotherapeutics [160,161,162,163]. Strategies to overcome myeloid-mediated suppression of the immune response can be generally categorized into (1) prevention of recruitment to the tumor, (2) depletion of myeloid populations or disruption of their suppressive mechanisms, and (3) repolarization towards immunostimulatory phenotypes. An overview of published studies targeting myeloid cells and their underlying strategies can be found in Table 2.

### 4.1. Disruption of Myeloid Recruitment

Both MDSC and TAM are recruited to the TME by chemoattractant molecules, so interference with binding of these molecules to their corresponding receptors can prevent the accumulation of myeloid cells close to the tumor and subsequently their immunoinhibitory and tumor-promoting effects. An interesting target is the CSF-1/CSF-1R axis, as it is involved in myeloid recruitment to the TME and differentiation into M2 phenotypes [164]. Indeed, studies in various solid malignancies have shown that blockade of the axis with monoclonal antibodies or CSF-1R inhibitors can lead to reduced myeloid numbers at the tumor site [165,166] and increased numbers of macrophages with antitumor activity [167]. Studying the spontaneous neuroblastoma TH-MYCN mouse model, Mao et al. have demonstrated that CSF-1R inhibition efficiently slows tumor progression in tumor-bearing mice and enhances PD-1/PD-L1 antibody therapy [160]. Furthermore, Webb et al. have shown that reduction of myeloid cells via CSF-1R blockade can also improve the efficiency of chemotherapy in a human neuroblastoma model in T cell-deficient mice, indicating a beneficial effect independent from the adaptive immune response [94]. The CCR2-CCL-2 axis is another pathway important in tumor-mediated myeloid recruitment. Albeit not investigated in neuroblastoma, studies in solid tumors show that the axis can be disrupted either by inhibiting CCR2 [168,169] or via anti-CCL2 antibodies [170,171,172], thereby preventing the recruitment of suppressive myeloid cells and initiation of their tumor-promoting effects. Furthermore, Carlson et al. have provided evidence that the tumor-associated inflammation in neuroblastoma can pose another therapeutic target to prevent the accumulation of suppressive effectors in the TME [7]. When treating TH-MYCN mice with low-dose aspirin, they reported a significant reduction in tumor burden, tumor-associated innate cells, including MDSC and TAM, and intra-tumoral TGF-β.

### 4.2. Depletion of Myeloid Populations

Alternatively, suppression exerted by myeloid cells in the TME can be counteracted either by disrupting their inhibitory mechanisms or depleting the myeloid populations entirely. Several agents have been reported to have cytotoxic effects on myeloid cells in solid tumors, among them chemotherapeutics, such as gemcitabine [173,174,175], 5-fluorouracil [176,177], and trabectedin [178,179]. The latter caused selective depletion of monocytes and macrophages in mouse models of fibrosarcoma, ovarian carcinoma, and Lewis lung carcinoma, as well as in patients with soft tissue sarcoma, while sparing lymphocytes and neutrophils [178]. Siebert et al., moreover, have reported beneficial effects of anti-CD11b antibody therapy in neuroblastoma-bearing mice in combination with anti-GD2 antibody therapy [180]. The receptor tyrosine kinase inhibitor sunitinib has also been suggested to induce myeloid depletion via interference with STAT3 signaling in several solid tumors, including renal cell carcinoma, melanoma, and fibrosarcoma [181,182,183,184]. Renal cell carcinoma patients treated with sunitinib have superior expansion of intratumor lymphocytes and decreased numbers of MDSC both in the tumor and in circulation [181,183]. In pediatric patients with solid malignancies, sunitinib treatment also results in decreased monocyte counts [185], however effects in neuroblastoma patients remain to be investigated.

One of the key inhibitory mechanisms of suppressive myeloid cells which has an established role in neuroblastoma is arginase activity [25]. Indeed, inhibition of arginase recovers T cell proliferation in vitro and reduces tumor growth in mouse models of colon, lung, skin, and breast cancer [186,187], and has shown beneficial effects in combination with checkpoint blockade, cell-based immunotherapy, and chemotherapy [186], indicating a potential for similar effects in neuroblastoma. The depletion of the essential amino acid tryptophan mediated by suppressive myeloid cells can also be overcome in various solid tumors through treatment with IDO inhibitors [188,189] and COX2 inhibitors [8,190,191].

### 4.3. Repolarization of Myeloid Cells

Suppressive myeloid cells can also be reprogrammed into their immunostimulatory counterparts by a range of therapeutics. These strategies typically function by blocking receptors inducing inhibitory signals or by providing exogenous ligands for receptors activating pro-inflammatory pathways [192]. One strategy to repolarize myeloid cells is targeting histone deacetylase (HDAC) proteins or epigenetic reader proteins [193,194]. One example for this is the HDAC inhibitor vorinostat, which has been shown to promote M1 repolarization of TAM as well as decreasing number and function of MDSC in an in vivo neuroblastoma study [195]. Kroesen et al. have further demonstrated that the drug mediates a synergistic effect with anti-GD2 monoclonal antibodies, making it an interesting candidate for combination immunotherapy. Similarly, HDAC inhibitor, entinostat, has been reported to suppress MDSC-mediated effects and enhance checkpoint inhibitor therapy in mouse models of breast cancer, pancreatic cancer, lung, and renal cell carcinoma [196,197]. Another agent described to induce the repolarization of myeloid cells is all-trans retinoic acid (ATRA), which upregulates expression of glutathione synthase in MDSC. The resulting accumulation of intracellular glutathione neutralizes their high ROS production and stimulates MDSC to differentiate into mature myeloid cells [198], which can improve immunotherapies, as seen with checkpoint inhibitors in melanoma patients [199] and cancer vaccines in patients with lung cancer [200,201]. Other approaches involving macrophage repolarization in neuroblastoma, including MSC-associated delivery of IFN-γ [202], inhibition of PGE2 [38], and viral delivery of CXCR4 antagonist [203], have all been shown to increase immunostimulatory M1 populations in the TME.

**Table 2 cancers-13-01743-t002:** Treatment strategies targeting myeloid cells in the tumor microenvironment. Table indicates way of action, therapeutics, the studied myeloid population, and references to publications. Asterisks indicate studies performed in neuroblastoma models.

Therapeutic Target	Point of Action	Treatment	Studied Population	Reference
CCR2/CCL2	Recruitment	CCR2 inhibitor	TAM/MDSC	[168,169]
		anti-CCL2 antibody		[170,171,172]
CSF-1/CSF-1R	Recruitment/ Repolarization	anti-CSF-1R antibody	TAM, MDSC	[94] *, [165,204,205,206]
		CSF-1R inhibitor	TAM, MDSC	[94] *, [160] *, [166,167,206,207,208,209]
RAGE	Recruitment	Anti-RAE antibody	MDSC	[128,210]
Epigenetics	Repolarization	BRD inhibitor	TAM	[193], [194] *
		Histone deacetylase inhibitor	TAM, MDSC	[195] *, [196,197]
Glutathione synthase (GSS)	Repolarization/ Survival	All-trans retinoic acid (ATRA)	MDSC	[162,198,199,200,201,211]
Syk-Rac2/PI3Kγ	Repolarization	Syk/PI3Kγ inhibitor	TAM, MDSC	[212,213]
CD40	Repolarization	CD40 agonist	TAM	[214] *, [215] *, [216] *, [217] *, [218]
PGE2	Repolarization	Prostaglandin E Synthase-1 inhibitor	TAM	[38] *
67 kDa laminin receptor	Repolarization	Polyphenon E	MDSC	[11] *
Bisphosphonates	Repolarization/ Survival	Liposome-encapsulated bisphosphonates, Zoledronic acid	TAM	[219,220,221,222,223]
JAK2/STAT3 and NF-κB	Repolarization/ Survival/ Function	JAK/STAT inhibitor	TAM	[72] *
		Sunitinib	MDSC	[181,182,183,184,224]
		Curcumin	MDSC	[225,226,227,228]
		Withaferin A	MDSC	[229]
Chemotherapy	Survival	5-fluorouracil (5-FU)	MDSC	[176,177], [180] *
		Gemcitabine	MDSC	[173,174,175,177,230]
		Trabectidin (ET-743)	TAM	[178], [179] *
CD11b	Survival	anti-CD11b antibody	MDSC	[180] *
CD1d	Survival	Vα24-invariant NKT cells	TAM	[19] *
CD105	Survival	anti-CD105 antibody	TAM	[231] *
CD38	Survival	Anti-CD38 antibody	M-MDSC/Tregs	[232,233,234]
Bcl-2/Bcl-xL	Survival	Bcl-2 inhibitor	MDSC	[235]
Folate receptor β (FRβ)	Survival	anti-FRβ CAR T cells	TAM	[161]
CD33/CD16	Survival	Bispecific killer cell enhancer (BiKE)	MDSC	[236]
		Gemtuzumab	MDSC	[163] *
TRAIL-R2	Survival	Agonist TRAIL-R2 antibody	MDSC	[237,238]
CD124 (IL-4Rα)	Survival	anti-IL4Rα aptamer	TAM/MDSC	[239]
CD47	Survival/Function	Anti-CD47 antibody	MDSC	[240,241]
Arg-1	Function	Arginase inhibitor	MDSC	[186,187]
IDO	Function/ Recruitment	IDO inhibitor	MDSC	[188,189]
COX2 (CXCR4/CXCL12)	Recruitment/ Function	COX2 inhibitor	MDSC	[7] *, [8] *, [190,191,242,243,244]
PDE-5	Function	PDE-5 inhibitor	MDSC	[245,246,247,248,249,250,251]
Checkpoint inhibitors	Function	Anti-PD-1 antibody Anti-CTLA-4 antibody	TAM/MDSC/ Tregs	[252,253]
NOX2	Function	NOX2 inhibitor	MDSC	[254]
VEGFR	Function	VEGFR antagonist	MDSC	[255]

## 5. Conclusions

In summary, myeloid populations in the TME of neuroblastoma accrue in a combined force, suppressing host immunity and immunotherapy, which needs to be considered when developing novel treatment approaches. Increasing numbers of myeloid cells within progressed lesions and the diagnostic value of myeloid markers in neuroblastoma underline the importance that these cellular players have in tumor progression and immune inhibition. Several studies have demonstrated that immunotherapy benefits from combinatorial approaches targeting myeloid-derived suppression alongside the tumor cells, and a combined effort will have to be put into finding the most effective combinations, which may change depending on therapeutic approach and disease stage. Further investigation may be required to more clearly dissect myeloid subpopulations and transitional stages to be able to pinpoint precise targets and predominant pathways when developing future interventions for neuroblastoma.

## Figures and Tables

**Figure 1 cancers-13-01743-f001:**
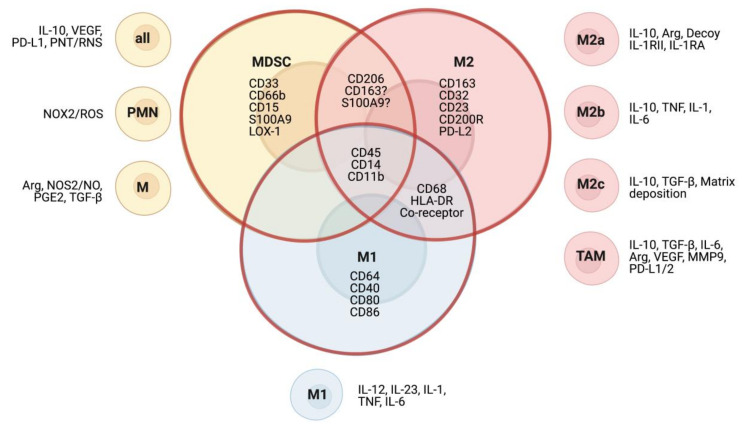
Phenotypic surface markers and functional subsets of myeloid cells in humans. Venn diagram shows typical cellular markers of MDSC, M1, and M2 macrophages. Markers shared between cell types are shown in overlapping areas of the diagram.

**Figure 2 cancers-13-01743-f002:**
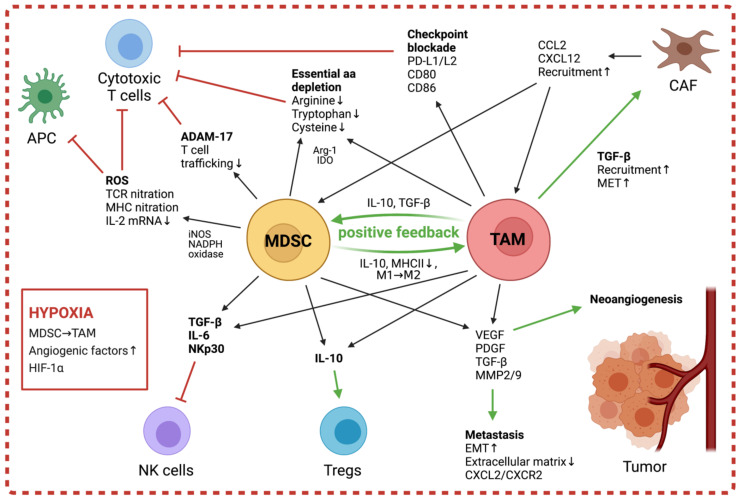
Suppressive mechanisms exerted by myeloid cells in the tumor microenvironment (TME) and their effects on its cellular components. Green arrows indicate stimulatory effects and red bar headed lines indicate inhibitory effects on targets.

**Table 1 cancers-13-01743-t001:** Subsets of myeloid cells in mouse and human. Table shows minimal markers for the phenotyping of subsets in macrophages and MDSC in mouse and human.

Cell Type	Mouse	Human
**Macrophages**	F4/80^+^ CD11b^+^ **M1:** CD80^high^ CD206^mid^ **M2/TAM:** CD80^mid^ CD206^high^	CD68^+^ CD11b^+^ CD14^+^ **M1:** HLA-DR^+^ CD80^+^ CD86^+^ **M2/TAM:** CD206^+^ CD163^+^
**MDSC**	Gr1^+^ CD11b^+^ MHC class II^low/−^ **PMN-MDSC:** CD11b^+^ Ly6C^lo^ Ly6G^+^ **M-MDSC:** CD11b^+^ Ly6C^hi^ Ly6G^−^	No general MDSC marker **PMN-MDSC:** CD14^−^ CD11b^+^ CD15^+^ (or CD66b^+^) LOX-1^+^ **M-MDSC:** CD11b^+^ CD14^+^ HLA-DR^lo/−^ CD15^−^ **e-MDSC:** Lin^−^(CD3/14/15/19/56) HLA-DR^−^ CD33^+^

## Data Availability

Not applicable.

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
