# Peer review of "Combined Effects of Myeloid Cells in the Neuroblastoma Tumor Microenvironment"

_cancers, 2021, doi:10.3390/cancers13071743_

Round 1

Reviewer 1 Report

The manuscript by Jennifer et al. entitled "Combined effects of myeloid cells in the neuroblastoma tumor microenvironment" provides an excellent review of how various myeloid cell lineages effect tumor development and growth, as well as some methods to manipulate these cells for better treatment outcomes. Although the manuscript overall is well written and provides an succinct comprehensive review, there are a few minor issues:

1) Lines 66-68: The sentence suggests that NK cells are with MHC and APMs, but these cells have other mechanisms of activation, which is mentioned in lines 70-72. But the differences in activation of NK vs T cells is not entirely clear.

2) Of particular concern, many aspects of aspects of tumor suppressive effects are given. However, the relative importance of each is not well defined. Are some cytokines and chemokines more or less important, especially when discussing neuroblastoma.

3) The authors correctly say "the binary M1 and M2 classification is an oversimplification" (line 181). However, the Figure 1 furthers the binary classification aspect of M1 and M2 cells. Can the discussions in lines 163-180 and 217-220 be incorporated into Fig 1?

4) Several sections are not neuroblastoma specific. Somewhere in these sections it should be stated that either the information is 1) not known for neuroblastoma or that 2) the authors think the information can likely be generalized to neuroblastoma. This is the only major criticism.

5) Of the pathways presented in Figure 2, which are thought to most important in neuroblastoma?

6) Line 443 suggests the discussion is about neuroblastoma patients, but it is not. Please clarify that the authors assume this can be generalized to neuroblastoma.

Minor editing

line 191: CD is defined here but used earlier.

It seems Fig 2 can be reference more often.

IDO is defined in line 360, but used earlier.

Line 351 should read "multitude of mechanisms"

In figure 2 legend, it references red arrows, but are they arrows?

Line 429: significantly should be significant.

Should "way of action" (table 2) be "mechanism of action"

Author Response

Reviewer 1

  • Lines 66-68: The sentence suggests that NK cells are with MHC and APMs, but these cells have other mechanisms of activation, which is mentioned in lines 70-72. But the differences in activation of NK vs T cells is not entirely clear.
    1. Deleted NK cells, as this effect only applies to T cells. Clarified that NK cells belong to innate immune responses mentioned in lined 73-74
  • Of particular concern, many aspects of aspects of tumor suppressive effects are given. However, the relative importance of each is not well defined. Are some cytokines and chemokines more or less important, especially when discussing neuroblastoma?
    1. This review focuses on the prevalent chemokines and cytokines in neuroblastoma, and from the literature review no hierarchy among them could be identified. Important soluble factors in myeloid-mediated suppression can be found in figure 2 and I added reference to Asgharzadeh et al. 2012 in lines 64-66, which provides a 14-gene signature for neuroblastoma, showing the importance of inflammation related genes (also listed in line 47-50).
  • The authors correctly say "the binary M1 and M2 classification is an oversimplification" (line 181). However, the Figure 1 furthers the binary classification aspect of M1 and M2 cells. Can the discussions in lines 163-180 and 217-220 be incorporated into Fig 1?
    1. Expanded figure 1 to include MDSC and macrophage subsets beyond M1 and M2 and included characteristic functions.
  • Several sections are not neuroblastoma specific. Somewhere in these sections it should be stated that either the information is 1) not known for neuroblastoma or that 2) the authors think the information can likely be generalized to neuroblastoma. This is the only major criticism.
    1. Clarified in the following lines by adding studied cancer types:
      1. Line 147
      2. Line 175
  • Line 220
  1. Line 271 -> deleted 'in neuroblastoma', because as pointed out this was first described in other cancers and only some of it has been shown in neuroblastoma
  2. Line 277-278
  3. Line 273
  • Line 279
  • Line 301-305
  1. Line 313-314
  2. Line 322-323
  3. Line 325 -> deleted 'in neuroblastoma', because as pointed out this was first described in other cancers and only some of it has been shown in neuroblastoma
  • Line 338-339
  • Line 360
  • Line 364-366
  1. Line 370-371
  • Line 381-382
  • Line 386
  • Line 416
  • Line 421-422
  1. Line 423-424
  • Line 426
  • Line 466
  • Line 476
  • Line 488
  • Line 497-501
  • Line 509-510
  • Line 511-512
  • Line 513-514
  • Line 535-536
  • Of the pathways presented in Figure 2, which are thought to most important in neuroblastoma?
    1. The answer to question 2 also applies to this question. Predominant pathways in neuroblastoma remain to be elucidated. Added this information in the conclusion, lines 566-567.
  • Line 443 suggests the discussion is about neuroblastoma patients, but it is not. Please clarify that the authors assume this can be generalized to neuroblastoma.
    1. Clarified that this was in RCC and added a source for paediatric tumors (lines 497-505)
  • Minor editing: all revised as requested

line 191: CD is defined here but used earlier. -> now in line 44

It seems Fig 2 can be reference more often. -> added in line 283, 296, 374, 395

IDO is defined in line 360, but used earlier. -> now in line 309

Line 351 should read "multitude of mechanisms" -> revised in line 396

In figure 2 legend, it references red arrows, but are they arrows? -> revised to say bar headed lines

Line 429: significantly should be significant. -> revised in line 482

Should "way of action" (table 2) be "mechanism of action") -> not really mechanisms, changed to point of action

Reviewer 2 Report

This is a well written and informative review that summarizes the current knowledge of the role of myeloid cells in the immunosuppressive microenvironment in neuroblastoma. I enjoyed reading the manuscript which should be of interest for researchers and clinicians focusing on neuroblastoma. I have only a few minor comments/recommendations for improvements (listed below) which should be easy to incorporate and should not impede accepting this review for publication.

MINOR COMMENTS:

  1. Throughout the manuscript, I have found sections where pieces of neuroblastoma-specific information merge with general knowledge or results obtained in different cancer types and sometimes it takes too much effort to distinguish whether the reference is neuroblastoma-related or it was used to put data into a broader perspective. Starting from chapter 2, the authors should try to better stress these bridges in the text and separate results obtained in neuroblastoma/neuroblastoma models from that obtained in other cancer types. This is important to avoid confusion of readers trying to understand the current state of knowledge in neuroblastoma.

    One example for all – 3.2.1. The Role of TAMs in Neuroblastoma – lines 247-251: Most references are neuroblastoma specific but are mixed with a general review on HIF (ref [96] which does not even mention neuroblastoma) without pointing that out. One can speculate if it was all discovered in neuroblastoma (as the heading would suggest) or if it is only a general introduction (as “neuroblastoma” is not mentioned anywhere in this part).
  1. In Table 2, it would be better to indicate studies performed in neuroblastoma models by placing the asterisks also right next to the number of the respective reference/references. This way it would be much easier for readers interested specifically in neuroblastoma to identify neuroblastoma-related studies.
  1. It would be helpful to provide a list of abbreviations.
  1. Line 103-105, the meaning of “TAMs in turn enhance STAT-3 activation by CAFs through” is not clear here: “…TAMs in turn enhance STAT-3 activation by CAFs through release of the soluble agonistic IL-6 receptor (IL-6R), and promote CAF invasiveness and proliferation [30], [36].” Was the intended meaning “STAT3 activation in CAFs”? Please also avoid the hyphen in “STAT-3” and use “STAT3” as in the rest of the manuscript.
  1. Line 171-172: “M1 and M2 macrophages do not only differentiate in their functions but can also be 171 distinguished by their genetic profile…”. I would recommend using “differ” instead of “differentiate” which might suggest a biological process to some readers.

Author Response

Reviewer 2

  • Throughout the manuscript, I have found sections where pieces of neuroblastoma-specific information merge with general knowledge or results obtained in different cancer types and sometimes it takes too much effort to distinguish whether the reference is neuroblastoma-related or it was used to put data into a broader perspective. Starting from chapter 2, the authors should try to better stress these bridges in the text and separate results obtained in neuroblastoma/neuroblastoma models from that obtained in other cancer types. This is important to avoid confusion of readers trying to understand the current state of knowledge in neuroblastoma.

One example for all – 3.2.1. The Role of TAMs in Neuroblastoma – lines 247-251: Most references are neuroblastoma specific but are mixed with a general review on HIF (ref [96] which does not even mention neuroblastoma) without pointing that out. One can speculate if it was all discovered in neuroblastoma (as the heading would suggest) or if it is only a general introduction (as “neuroblastoma” is not mentioned anywhere in this part).

  1. Clarified in the following lines by adding studied cancer types:
    1. Line 147
    2. Line 175
  • Line 220
  1. Line 271 -> deleted 'in neuroblastoma', because as pointed out this was first described in other cancers and only some of it has been shown in neuroblastoma
  2. Line 277-278
  3. Line 273
  • Line 279
  • Line 301-305
  1. Line 313-314
  2. Line 322-323
  3. Line 325 -> deleted 'in neuroblastoma', because as pointed out this was first described in other cancers and only some of it has been shown in neuroblastoma
  • Line 338-339
  • Line 360
  • Line 364-366
  1. Line 370-371
  • Line 381-382
  • Line 386
  • Line 416
  • Line 421-422
  1. Line 423-424
  • Line 426
  • Line 466
  • Line 476
  • Line 488
  • Line 497-501
  • Line 509-510
  • Line 511-512
  • Line 513-514
  • Line 535-536
  • In Table 2, it would be better to indicate studies performed in neuroblastoma models by placing the asterisks also right next to the number of the respective reference/references. This way it would be much easier for readers interested specifically in neuroblastoma to identify neuroblastoma-related studies.
    1. Moved asterisks to 'Reference' column to the neuroblastoma-related studies in Table 2 and deleted asterisks in 'Treatment' column.
  • It would be helpful to provide a list of abbreviations.
    1. List of abbreviations at the end of this file and in Appendix A, line 575.
  • Line 103-105, the meaning of “TAMs in turn enhance STAT-3 activation by CAFs through” is not clear here: “…TAMs in turn enhance STAT-3 activation by CAFs through release of the soluble agonistic IL-6 receptor (IL-6R), and promote CAF invasiveness and proliferation [30], [36].” Was the intended meaning “STAT3 activation in CAFs”? Please also avoid the hyphen in “STAT-3” and use “STAT3” as in the rest of the manuscript.
    1. Agonistic soluble IL-6 receptor from TAMs binds released IL-6 from CAFs, resulting in enhanced presentation of the cytokine to gp130 and the subsequent activation of STAT3 in the target cells, for example neuroblastoma cells. Clarified that CAF-mediated STAT3 activation is meant (line 121).
  • Line 171-172: “M1 and M2 macrophages do not only differentiate in their functions but can also be 171 distinguished by their genetic profile…”. I would recommend using “differ” instead of “differentiate” which might suggest a biological process to some readers.
    1. Changed to 'differ' (line 193)

Reviewer 3 Report

Title: Combined effects of myeloid cells  in the neuroblastoma tumor microenvironment.

In this manuscript the authors give a comprehensive review on a relatively new domain: the role of the cellular components of the microenvironment of (neuroblastoma) tumors in the processes of immunosuppression and tumor proliferation. It summarizes how the inflammation characteristic for neuroblastoma induces suppressive myeloid cells in tumor microenvironment. The myeloid cell population in the tumor microenvironment consists of tumor associated macrophages (TAMs) and of myeloid derived suppressor cells (MDSCs) and correlates with advanced neuroblastoma tumors and poor outcome. An overview is given of the different processes involved in immunosuppression and proliferation and immune evasion of tumor cells based on extensive data from literature.  Such knowledge is important in development of new clinical trials in an attempt to overcome myeloid-mediated suppression in neuroblastoma. Indeed, clinical trials focusing on tumor elimination by activation and/or modification of host immunity e.g CAR T therapy in neuroblastoma failed to reproduce the results obtained in preclinical studies. Knowledge of the mechanisms of induction and suppression of the tumor microenvironment associated myeloid cell population is important in the design of novel immunotherapeutic treatments, not only in  neuroblastoma but also for other solid tumors.

Strengths:

  • This is an excellent review on a relatively new subject in neuroblastoma, the tumor microenvironment as a major contributor to immune suppression and tumor evasion.
  • This information may guide the development of clinical trials and therapies relying on functional immune response and/or combination therapies
  • The manuscript is well written, the English language is appropriate and understandable.

(One typo: environment, instead of enviornment line 133.

Also line 171: M1 and M2 do not only differentiate in their functions: differ? )

  • There is an exhaustive list of references

Author Response

Reviewer 3

  • One typo: environment, instead of enviornment line 133.
    1. Revised in line 151
  • Also line 171: M1 and M2 do not only differentiate in their functions: differ?
    1. changed to 'differ (line 193)

Additional changes:

Added 'Acknowledgements: All figures were created with BioRender.com.' (line 573)

Line 434: deleted 'Toll-like receptor', abbreviation was defined before

Line 196: added reference to figure 1

Line 327: deleted 'NB' as abbreviation was never introduced